# Mode of Action of Toxin 6-Hydroxydopamine in SH-SY5Y Using NMR Metabolomics

**DOI:** 10.3390/molecules30163352

**Published:** 2025-08-12

**Authors:** Roktima Tamuli, George D. Mellick, Horst Joachim Schirra, Yunjiang Feng

**Affiliations:** 1Institute for Biomedicine and Glycomics, Griffith University, Nathan, QLD 4111, Australia; roktima.tamuli@griffithuni.edu.au (R.T.); g.mellick@griffith.edu.au (G.D.M.); h.schirra@griffith.edu.au (H.J.S.); 2School of Environment and Science, Griffith University, Nathan, QLD 4111, Australia; 3Centre for Advanced Imaging, The University of Queensland, Brisbane, QLD 4072, Australia

**Keywords:** 6-OHDA, NMR metabolomics, SH-SY5Y, ETC, ROS

## Abstract

This study used NMR-based metabolomics to investigate the mode of action (MoA) of 6-hydroxydopamine (6-OHDA) toxicity in the SH-SY5Y neuroblastoma cell model. 6-OHDA, a structural analogue of dopamine, has been used to create a Parkinson’s disease model since 1968. Its selective uptake via catecholaminergic transporters leads to intracellular oxidative stress and mitochondrial dysfunction. SH-SY5Y cells were treated with 6-OHDA at its IC_50_ concentration of 60 μM, and samples of treated and untreated groups were collected after 24 h. The endo metabolome was extracted using a methanol–water mixture, while the exo metabolome was represented by the culture media. Further, endo- and exo metabolomes of treated and untreated cells were analysed for metabolic changes. Our results demonstrated significantly high levels of glutathione, acetate, propionate, and NAD^+^, which are oxidative stress markers, enhanced due to ROS production in the system. In addition, alteration of myoinositol, taurine, and o-phosphocholine could be due to oxidative stress-induced membrane potential disturbance. Mitochondrial complex I inhibition causes electron transport chain (ETC) dysfunction. Changes in key metabolites of glycolysis and energy metabolism, such as glucose, pyruvate, lactate, creatine, creatine phosphate, glycine, and methionine, respectively, demonstrated ETC dysfunction. We also identified changes in amino acids such as glutamine, glutamate, and proline, followed by nucleotide metabolism such as uridine and uridine monophosphate levels, which were decreased in the treated group.

## 1. Introduction

Parkinson’s disease (PD), the second most common neurodegenerative disorder affecting the global population aged 60 years and older, leads to the loss of dopaminergic neurons in substantia nigra par compacta [1,2]. 6-hydroxydopamine (6-OHDA), an oxidisable hydroxylated analogue of dopamine, has a high affinity for dopamine transporters [3], leading to the selective damage of tyrosine hydroxylase-containing neurons, i.e., dopaminergic neurons of substantia nigra par compacta [4,5]. It can decrease dopaminergic neuron populations by over 60% [5] and is popularly used to mimic PD in biological models [3,4,6,7]. 6-OHDA-induced in vitro and in vivo PD models are among the most extensively utilised research models for understanding the neuropathological and biochemical mechanisms underlying PD, as well as for evaluating potential therapeutic strategies [3,4,6,7,8,9].

6-OHDA-induced Parkinsonian effects have been extensively studied in different animal models [6,7] and cell models [10,11], revealing its mechanism of toxicity involves reactive oxygen species (ROS)-induced oxidative stress and mitochondrial dysfunction [6,12,13]. Additionally, intracellular oxidative stress triggered by 6-OHDA has been reported to interact with α- synuclein, a key protein associated with PD pathology [5]. Lipidomic analysis of the 6-OHDA-treated SH-SY5Y cells displayed key lipid alterations that closely resemble those observed in PD patient brains [14].

However, a review of the literature reveals NMR metabolomics has not yet been employed to elucidate MoA of 6-OHDA, which formed the goal of our study. This work serves as a proof-of-concept to validate the use of NMR metabolomics in this cell model when treated with the toxin. SH-SY5Y cells are a subline of the SK-N-SH neuroblastoma cell line, which displays catecholaminergic neuronal properties, including the expression of tyrosine hydroxylase and dopamine beta hydroxylase enzymes [15,16]. Furthermore, these cells hold the capacity to synthesise both dopamine and noradrenaline neurotransmitters [15]. Given that the degeneration of dopaminergic neurons is a hallmark of PD, and SH-SY5Y cells display the dopaminergic phenotype, this is a good cell model for use in PD research [16].

MoA, a series of complex multi-layered procedures, can be defined by the connection between biomolecular interactions and the translation of these events to phenotypic changes [17]. Decoding the MoA provides insight into the molecular and biochemical knowledge of the treatment of interest.

NMR-based metabolomics [18,19] is a non-destructive method that has proved to be an effective tool in different areas of research, such as biomarker identification [20,21], pathogenesis of disease [21,22], environmental research [21], plant metabolomics [23,24], and quality and quantity of metabolites in food [25], including mode of action (MoA) studies [18,26]. Complete metabolome analysis comprises endo and exo metabolome analysis. The endo metabolome encompasses all intracellular metabolites, providing comprehensive knowledge of the metabolic events that navigate biochemical and physiological processes, as well as interpreting the cell’s response to different signals [27,28,29]. Furthermore, the exo metabolome aids in the analysis of metabolites that cells uptake or excrete into and from the culture media [27,28].

This study utilised SH-SY5Y cells treated with 6-hydroxydopamine at its IC_50_ concentration (60 μM) to model Parkinson’s disease (PD). Both intracellular (endo metabolome) and extracellular (exo metabolome) metabolites from treated and control groups were analysed using NMR spectroscopy. Comprehensive multivariate statistical analyses revealed significant metabolic alterations, highlighting the biochemical pathways affected by 6-OHDA. These findings contribute to a deeper understanding of 6-OHDA’s effects in this cell model commonly used in PD research, using the specialised technique, NMR metabolomics.

## 2. Results

### 2.1. Multivariate Statistical Analysis of the Metabolic Profiles of Endo Metabolome

SH-SH5Y cells were cultured to 75–85% confluency, followed by treatment with 60 μM 6-OHDA for 24 h. Harvested cells were pelleted down and the endo metabolome was extracted with methanol and water. A total of 34 endo metabolites were identified (Appendix A). Multivariate statistical analysis on the 1D NMR spectra was performed. The principal component analysis (PCA) based on *n* = 28 spectra and *k* = 9010 variables yielded a model with five principal components (PCs), a total explained variance of *R*^2^ = 0.725, and a cross-validated predictive *Q*^2^ value of 0.412. The scores plot of PC1 and PC2 (Figure 1a) shows separation in the PC1 (horizontal) axis between the treated (red) and untreated (blue) groups. No missing data were found in the PCA models obtained. One outlier is noticeable outside Hotelling’s 95% confidence range in the PCA model. Metabolites extracted from the cells cultured with and without 6-OHDA showed distinct separation.

Thirty-four metabolites of the endo metabolome were identified in the bivariate loadings plot (Figure 1b). Among these, 5 metabolites had significantly higher levels in the treated cells compared to the controls and 14 were observed to have lower levels (Table 1). The levels of 15 metabolites remained unchanged by the treatment (Appendix A). Additional metabolomic analysis of four separate biological replicates/group, including three technical replicates, was performed, further validating our data (Appendix A).

An unpaired *t*-test followed by Benjamini Hochberg correction was used to check if the treated and untreated groups were different in terms of their mean values (Table 1). The peak integration area of a non-overlapping ^1^H NMR signal for each metabolite was used to perform the unpaired *t*-test (*n* = 17 in both groups). The mean values of both groups were used to determine the fold change ratio. The total metabolite content in each sample was estimated by calculating the total spectral intensity across the NMR spectrum (Appendix A). This analysis indicated that the total metabolite content in the treated endo metabolome was lower than that of the untreated group, indicating a lower number of viable cells in the treated group due to cell death. Therefore, normalisation of the endo metabolome levels was performed based on the total metabolite content of viable cells.

### 2.2. Multivariate Statistical Analysis of the Metabolic Profiles of Exo Metabolome

Principal component analysis was performed on the exo metabolome similar to the endo metabolome analysis. The PCA based on *n* = 31 spectra and *k* = 9520 variables yielded a model with 8 PCs, *R*^2^ = 0.86, and *Q*^2^ = 0.654. The PC1/PC2 scores plot of the PCA analysis (Figure 2a) showed separation in the PC1 (horizontal) axis between the treated (red) and untreated (blue) groups. No outliers or missing data were observed. This highlights the differences in metabolic profiles between both groups. Metabolites contributing to this difference were then identified by generating a bivariate loadings plot for PC1.

The exo metabolome bivariate loadings plot (Figure 2b) led to the identification of 20 metabolites. Among these, nine metabolites showed no change between the treatment groups (Appendix A), whereas seven metabolites, such as leucine, valine, isoleucine, acetate, alanine, formate, and glutamine, had significantly higher levels in the treated group. Lactate, pyruvate, glutamate, and pyroglutamyl alanine were four metabolites that exhibited lower levels in the 6-OHDA-treated group compared to the untreated. An unpaired *t*-test followed by Benjamini Hochberg correction was performed to determine the *p*-value and fold difference (Table 2) between both groups. Peak integration area of a non-overlapping ^1^H NMR signal for each metabolite has been utilised to perform the unpaired *t*-test. The number of samples in both groups was 19. The total metabolite content was estimated by calculating the total spectral intensity across the NMR spectrum, using the same approach as for the endo metabolome (Appendix A). This analysis showed that the total metabolite content in the exo metabolome was not significantly affected by cell death; therefore, non-normalised data were used for subsequent analyses.

### 2.3. Altered Metabolic Pathways

Metabolic pathway analysis was conducted using MetaboAnalyst 6.0 [30] to identify the altered metabolic pathways of endo and exo metabolome following 6-OHDA treatment. Seven different pathways have been identified in the endo metabolome of 6-OHDA treated cells, namely: glutathione metabolism, taurine and hypotaurine metabolism, pyrimidine metabolism, pyruvate metabolism, galactose metabolism, glycolysis and gluconeogenesis, arginine and proline metabolism, and glyoxylate and dicarboxylate metabolism (Figure 3). The affected pathways in the exo metabolome of treated cells were arginine biosynthesis; pyruvate metabolism; glycolysis and gluconeogenesis; alanine; aspartate and glutamate metabolism; valine, leucine, and isoleucine metabolism; and glyoxylate and dicarboxylate metabolism (Figure 4).

### 2.4. Pyroglutamyl Alanine

Pyroglutamyl alanine is one of the metabolites whose levels are significantly higher in the exo metabolome of untreated cells (Figure 2b). The structure of pyroglutamyl alanine was elucidated using 1D and 2D NMR (Appendix A). Under normal conditions, pyroglutamyl alanine is expected to break down into alanine and glutamate. We noticed the breakdown of pyroglutamyl alanine into alanine after 24 h incubation with or without cells. However, when cells were treated with 6-OHDA, breakdown into alanine is even higher (Figure 5). The fate of this metabolite is not clear from our study and remains inconclusive. Pyroglutamyl alanine is present in cell culture media, especially in mammalian cell culture media, where it is added as a bioactive supplement to enhance production and cellular proliferation [31].

## 3. Discussion

In this study, NMR metabolomics was used as a tool to understand the MoA of 6-OHDA in the SH-SY5Y cell line. It is an attempt to correlate the established mechanism of toxicity of 6-OHDA with our data. In doing so, we were able to identify and analyse metabolites that were significantly altered due to the treatment. Understanding those metabolic changes led us to a mechanism centred on defective mitochondrial electron transport chain and an ROS-induced oxidative stress-related response. A schematic representation (Figure 6) of the pathways impacted has been represented based on MetaboAnalyst 6.0 web server and traditional biochemical knowledge.

### 3.1. Reactive Oxygen Species

The toxicity of 6-OHDA is resulted, in part, from the formation of reactive oxygen species (ROS) leading to oxidative stress [9,11]. 6-OHDA, a substrate of monoamine oxidase [32], quickly and non-enzymatically undergoes auto-oxidation to generate toxic species such as quinones, superoxide radicals, and hydrogen peroxide, including hydroxyl radicals [13]. ETC dysfunction due to complex I inhibition would also lead to ROS production.

Glutathione (GSH) (a tripeptide of L-γ-glutamyl-L-cysteinyl-glycine) is known to scavenge free radicals and is an important intracellular thiol/sulphydryl compound found in mammalian cells [33,34,35,36]. In our study, increased levels of GSH in the endo metabolome of the treated cells indicate the presence of oxidative stress and ROS. Decreased levels of NAD^+^ indicate the presence of oxidative stress. PARP-mediated NAD^+^ depletion confirms the phenomena [37]. Glucose-derived pyruvate has been converted to acetate in the presence of ROS [38], which explains the high level of acetate in the system and low level of pyruvate. A significant decrease in o-phosphocholine, myoinositol, and taurine in the treated group could indicate a change in membrane permeability due to abnormal membrane potential and osmolarity [39] induced by ROS. The decrease in myoinositol could also be explained due to the lack of glucose-6-phosphate in the treated cell, as glucose-6-phosphate converts myoinositol-1-phosphate to myoinositol [40].

### 3.2. Electron Transport Chain Dysfunction

Complex I inhibition of the mitochondrial electron transport chain is one of the proposed MoA of 6-OHDA [9,11], leading to a decrease in mitochondrial respiration and ATP depletion [12,41]. Short-chain fatty acids, namely acetate and propionate, in the presence of ROS involve β-oxidation shifting the energy production from glycolysis to fatty acid metabolism, leading to a high level of glucose in the system. This indicates that glycolysis has been affected due to ETC dysfunction. An additional group of metabolites such as creatine, creatine phosphate, glycine, and methionine are the consequences of ATP depletion inside the cell. Creatine maintains the cellular energetics [39] and converts to creatine phosphate utilising ATP. Presence of ROS would lead to oxidative stress which would result in decreased creatine levels. Creatine is formed by glycine in the presence of methionine. This metabolite could be traced back to acetyl CoA, as acetyl CoA forms lipids, which convert into choline and choline in turn generates glycine. Alanine, another product of the metabolism of pyruvate, or formed due to the breakdown of pyroglutamyl alanine, has been secreted into the extracellular environment. Increased alanine also explains the low level of lactate as pyruvate might have converted to alanine instead of lactate. The TCA cycle being a crucial step in energy metabolism forms 2-oxoglutarate [42] which further converts to glutamate [39] leading to amino acid metabolism and nucleotide metabolism. Both these pathways are affected in the 6-OHDA treated cells; proline [42,43], glutamate, glutamine, uridine, and uridine monophosphate decreased in treated endo metabolome along with decreased glutamate level in exo metabolome too.

### 3.3. Other Research

Mitochondrial complex I (NADH dehydrogenase) plays a crucial role in ATP generation [44,45] which maintains the NAD^+^ pool and protonmotive force (PMF) [46]. Complex I gained attention because of its role in neurodegenerative diseases. Inhibition of complex I results in the depletion of ATP and the disruption of PMF; both of these effects have been identified in our study. Single electron leakage from complex I can combine with oxygen to form superoxide anions, and enhanced production of ROS has been previously reported in complex I dysfunction [32]. This supports our work where inhibition of complex I by 6-OHDA escalated oxidative stress due to ROS.

Previous research has grouped known complex I inhibitors into different classes: class A and class B, based on their ability to produce ROS [32]. Class A inhibitors include rotenone, piericidin A, and rolliniastatin-1 and -2, and class B inhibitors include stigmatellin, capsaicin, mucidin, and coenzyme Q2 [32,44]. However, based on the effects of the kinetic behaviour of the enzyme, complex I inhibitors can be grouped into three classes. Class I/A includes piericidin A as the prototype, Class II/B includes rotenone, and Class C includes capsaicin as the prototype [32,44]. 6-OHDA, based on the ability to produce ROS, would be a class A inhibitor. It is a catecholaminergic [12] toxic oxidative form of dopamine [8] that exhibits structural similarities to catecholamines, dopamine, and noradrenaline [6]. 6-OHDA to date has been reported to have three mechanisms for its toxicity: (1) production of hydrogen peroxide, superoxide, and hydroxyl radical due to auto-oxidation of 6-OHDA intra or extra-cellularly; (2) hydrogen peroxide enhancement as a result of monoamine oxidase; and (3) inhibition of mitochondrial complex I causing ETC dysfunction [6,8,10,11,12,13]. Processes (1) and (2) occur independently or could be enhanced by inhibition of complex I, leading to ROS and oxidative stress. Our study showed the consequences of ROS-induced metabolites, which reflect the already established MoA of 6-OHDA. We further noticed the consequences of ATP depletion due to ETC dysfunction, leading to defects in energy metabolism, amino acid metabolism, glycolysis, and nucleotide metabolism.

### 3.4. Limitations and Future Directions

One of the key limitations of this study is the use of a single concentration of 6-OHDA, which restricts our capacity to acquire the concentration-dependent effects of the toxin. Understanding the metabolic profile across different 6-OHDA concentrations would help gain more comprehensive knowledge of its dose-dependent impact on biochemical pathways, including energy metabolism, oxidative stress, and mitochondrial dysfunction. Such an approach could display thresholds for toxicity, identify early metabolic biomarkers of cellular stress, and help specify changes that occur with increasing toxin exposure. Future studies should incorporate a range of 6-OHDA concentrations to refine our understanding of the molecular events that drive the degeneration of dopaminergic neurons. Another promising direction for future research is analysing the metabolic profile of isolated mitochondria from a 6-OHDA-injected animal model. Such an approach could complement and validate our current findings while highlighting key differences between cell-based and animal model analysis. As mitochondria play a major role in cellular energy metabolism and are critical regulators of many fundamental biological processes [47], studying their isolated metabolic activity would provide targeted insights into the mitochondrial-specific effects of 6-OHDA. This could enhance our understanding of organism-level metabolic alterations and offer a more integrated view of Parkinson’s disease pathophysiology.

## 4. Materials and Methods

### 4.1. Cell Culture

SH-SY5Y cells were cultured in a T25 flask in DMEM/F12 media supplemented with 10% FBS, GlutaMax, and non-essential amino acids in a humidified incubator with 5% CO_2_ in air at 37 °C. Initially, 2.5 × 10^4^ cells/well of SH-SY5Y cells were seeded in a 96-well plate. At 2.5 × 10^4^ cells/well, SH-SY5Y was not up to 70% confluent until after 48 h; hence, cell density was doubled to 5.0 × 10^4^ cells/well, which became 85% confluent after 24 h. Hence, 5.0 × 10^4^ cells/well of SH-SY5Y cells were determined to be the best seeding densities in a 96-well plate.

### 4.2. IC_50_ Determination of 6-OHDA

To obtain the IC_50_ and generate a dose–response curve for 6-OHDA, an MTT assay was carried out at the optimised cell density of 5.0 × 10^4^ cells/well. The stock solution of 6-OHDA was prepared at 30 mM in H_2_O containing 0.1% (*v*/*v*) sodium metabisulphite. As shown in Figure 7, 6-OHDA has 100% efficacy, although its potency is suboptimal with IC_50_ 54.01 μM (Figure 5). The working concentration was 60 μM to maintain a cell viability range of 50–70% and to allow for scalability during large-scale experiments.

### 4.3. Treatment of Cells for NMR Metabolomics

SH-SY5Y cells were cultured in T25 flasks and maintained in 5% CO_2_, 37 °C condition until 75–85% confluent. Three biological repeats (three different passages) and six technical repeats (six different sets of cultures) were used in NMR metabolomics experiments. Fresh media without 6-OHDA was added to the control group, and 60 μM of 6-OHDA was added to the treatment group, followed by incubation for 24 h. At the end of the incubation period, 1 mL of culture media from each flask was collected for exo metabolome analysis. The collected medium was immediately stored at −80 °C for future analysis. The remaining medium was aspirated, followed by PBS wash to remove any cell debris and trypsinisation. Harvested cells were centrifuged at 0.2 rcf for 5 min, discarding the supernatant, the pellets were placed on ice and subjected to metabolite extraction. A total of 72 samples, 36 treated (18 exo and 18 endo) and 36 untreated (18 exo and 18 endo), were prepared and used for NMR experiments, including four pooled quality control samples (PQC).

### 4.4. Intracellular Metabolite Extraction

The intracellular metabolites were extracted according to the extraction protocol by Bhinderwala et al. [26]; 1 mL of −20 °C pre-chilled methanol was added to the collected cell pellets to quench and lyse cells. The tubes were then centrifuged (Micro 185 centrifuge, Tuttlingen, Germany) at 13,000× *g* for 5 min at 4 °C. Supernatant was collected in a new tube. For the second extraction, 0.5 mL of 8:2 methanol/water was added to the pellet and centrifuged for 5 min. All experiments were carried out at 4 °C. Supernatant was combined with the previously collected extract. The third extraction was performed by adding 0.5 mL water and centrifuging at 13,000× *g* for 5 min. All combined supernatants were lyophilised using a Christ Freeze-dryer (Osterode am Harz, Germany), and dried cell extract was stored at −80 °C for future analysis. For analysis of samples, two different buffers were prepared. A 1.5 M potassium phosphate buffer was prepared for cell extracts. Phosphate buffer was prepared by combining dipotassium hydrogen phosphate (K_2_HPO_4_) and potassium dihydrogen phosphate (KH_2_PO_4_) salts and adjusting to pH 7.4. For analysing media samples, 0.09 M sodium phosphate buffer was prepared by combining disodium phosphate (Na_2_HPO_4_) and sodium phosphate monobasic (NaH_2_PO_4_) salts at pH 7.4. Both the buffers were prepared in sterile H_2_O and stored at 4 °C.

### 4.5. NMR Sample Preparation of Endo and Exo Metabolome

NMR samples of cell extracts were prepared by dissolving the dried extract in 200 μL of miliQ water followed by centrifuging (Micro 185 centrifuge, Tuttlingen, Germany) at 13,000× *g* for 5 min. Then, 180 μL supernatant was collected, and 50 μL of potassium buffer, pH 7.4, was added as well as 50 μL of a solution of 1 mM 1,1-difluoro-1-trimethylsilanyl methyl phosphonic acid (DFTPM, internal pH standard) containing 1 mM 2,2-dimethyl-2-sila-3,3,4,4,5,5-hexadeuteropentane-5-sulfonic acid (DSS, chemical shift standard) in D_2_O, followed by centrifugation at 13,000× *g* for 10 min. Subsequently, 200 μL supernatant was taken for NMR analysis, and 20 μL was taken to create pooled quality control (PQC) samples. For NMR sample preparation of media, the frozen medium was thawed and centrifuged to avoid any cell debris. 150 μL of supernatant was added with 80 μL of sodium buffer and 20 μL of D_2_O. Then, 200 μL was taken for NMR analysis, and 20 μL was taken for PQC creation. All the sample preparation steps were carried out at 4 °C to avoid any degradation of metabolites or cross-reaction.

### 4.6. NMR Measurements

NMR samples were prepared in a 3 mm tube and placed in a 96-well sample carrier in a randomised manner. The randomisation of samples has been conducted by a stratified permuted block randomisation technique [48]. All experiments were performed at 27 °C with automatic tuning matching at the operating ^1^H frequency 800.25 MHz in a Bruker 800 MHz spectrometer (Bruker Biospin, Rheinstetten, Germany), equipped with a cryo-probe with *z*-axis gradient coil and a 4 °C sample storage facility. The 90° pulse was adjusted to ~8 μs. For cell extracts, 1D-NOESY NMR data were obtained by collecting a total of 256 scans into 64 k data points with a spectral width of 20 ppm. For media, data were acquired by collecting 32 scans into 98 k data points with a spectral width of 30 ppm. CPMG spectra of the medium were obtained at 128 scans into 72 k data points with 20 ppm spectral width. Water suppression was achieved by low power continuous irradiation of the water resonance during the relaxation delay d1 of 4.00 s. For all 36 samples, 2D ^1^H-^1^H J-resolved spectra were also recorded. In addition, COSY, TOCSY, HSQC-TOCSY, and HMBC were acquired on one PQC sample of endo metabolome and one PQC sample of exo metabolome to facilitate metabolite identification.

### 4.7. NMR Data Preprocessing

All NMR spectra were processed in Bruker TopSpin 4.3 (Bruker Biospin, Rheinsttetten, Germany) by multiplication with an exponential function with a line broadening equivalent to 0.30 Hz Fourier Transform followed by manual calibration, phase correction, and baseline correction using Bruker TopSpin 4.3. For cell extracts, the spectra were calibrated at 0.0 ppm for DSS. For cell media, spectra were calibrated at the anomeric doublet of glucose at 5.22 ppm. Phase correction involved adjusting both the zero and first-order phases.

Correction of pH and ionic-dependent shifts of NMR signals was conducted with the icoshift algorithm in MATLAB 9.14.0 (MathWorks, Natick, MA, USA). This step was followed by bucketing of NMR spectra to reduce data with an in-house MATLAB script to consecutive integral regions of 0.001 ppm width (“buckets”), covering the range of δ = 10.0–0.25 ppm. The chemical shift regions at δ = 5.20–4.37 ppm and δ = 5.40–4.12 ppm were excluded for exo and endo metabolome, respectively, to eliminate the water signal. The bucketed integrals were normalised to the total intensity of the spectrum to correct any sample differences in weight and dilution. Normalised bucketed data matrices were then imported to the SIMCA 16 software package (Sartorius Stedim AB, Umeå, Sweden) for multivariate statistical analysis.

### 4.8. Multivariate Analysis

Pareto-scaled bucketed 1D spectra were taken for multivariate analysis. Principal component analysis (PCA) was performed to understand the differences in both groups and determine any outliers. A single outlier was identified in the endo metabolome PCA model, which we decided to retain as it followed the same trend in PC1 as the others. Elevated and decreased metabolite levels were determined from a bivariate 1D loadings plot, where loadings coefficients *p* were plotted against the chemical shifts of the metabolites. The absolute values of the correlation scaled loadings coefficients |*p*(*corr*)| were superimposed on the loadings plot as a heatmap colour scale. Loadings plots were generated using an in-house MATLAB script. Spectral features representing significantly altered metabolites were identified from the 1D loadings plots, in which loadings coefficients *p* were plotted against chemical shifts (ppm) of the metabolites and correlation scaled loadings coefficients |*p*(*corr*)| were superimposed on the loadings plot as a heatmap colour scale [49,50]. A bivariate loadings plot provides the same information as a traditional S-plot, allowing identification of variables with high or low significance.

### 4.9. Metabolite Identification

Metabolite signals were assigned using Chenomx NMR Suite v.9 and the online databases Human Metabolome Database and Biological Magnetic Resonance Bank. To aid in this, assignment of metabolites was also performed by COSY, TOCSY, HSQC-TOCSY, and HMBC. These spectra were also considered in the identification of unknown metabolites or anything not available in the databases. The covariance matrix (STOCSY) [51] at full spectral resolution was calculated to identify and solve the structure of pyroglutamyl alanine.

### 4.10. Metabolite Integration and Statistical Analysis

Subsequently, an in-house MATLAB script was used to integrate isolated peaks of each metabolite. These peak intensities were submitted to Prism v10.4 (GraphPad Software, Boston, MA, USA) to perform unpaired *t*-test analysis, which determined the *p*-values for each metabolite followed by Benjamini Hochberg correction. The means of the treated and untreated groups were used to indicate the fold change in the increase or decrease in the metabolites. The endo metabolome levels were normalised to the total metabolite content of spectra, as analysis indicated that the total metabolite content in the treated endo metabolome was lower than that of the untreated group, indicating a lower number of viable cells in the treated group due to cell death. Metabolic pathway analysis was performed with MetaboAnalyst 6.0 [30]. Lists of those identified metabolites which significantly changed in the endo or exo metabolome, respectively, were entered. The *Homo sapiens* (KEGG) library was selected for metabolic pathway analysis. The algorithms used for over-representation and pathway topology analyses were the hypergeometric test and relative-betweeness centrality, respectively. Pathways were considered significant when the *p*-values calculated from the enrichment analysis were less than 0.05.

## 5. Conclusions

In this study, we demonstrated metabolic disturbances induced by 6-OHDA in SH-SY5Y cells, mainly through ROS-generated oxidative stress and mitochondrial dysfunction. Our data indicated the consequences of potential ATP depletion, which is reflected in reduced glycolysis and downstream impairments in energy and amino acid metabolism. The absence of detectable ATP in our samples might be due to the instability under varying pH, time, and temperature conditions. Rather than identifying a single metabolite driving these changes, we were able to map a broader pathological mechanism involving ETC dysfunction, causing a cycle of ROS-induced oxidative stress. These insights validate the utility of NMR metabolomics in exploring future studies involving different PD models in a dose-dependent manner.

## Figures and Tables

**Figure 1 molecules-30-03352-f001:**
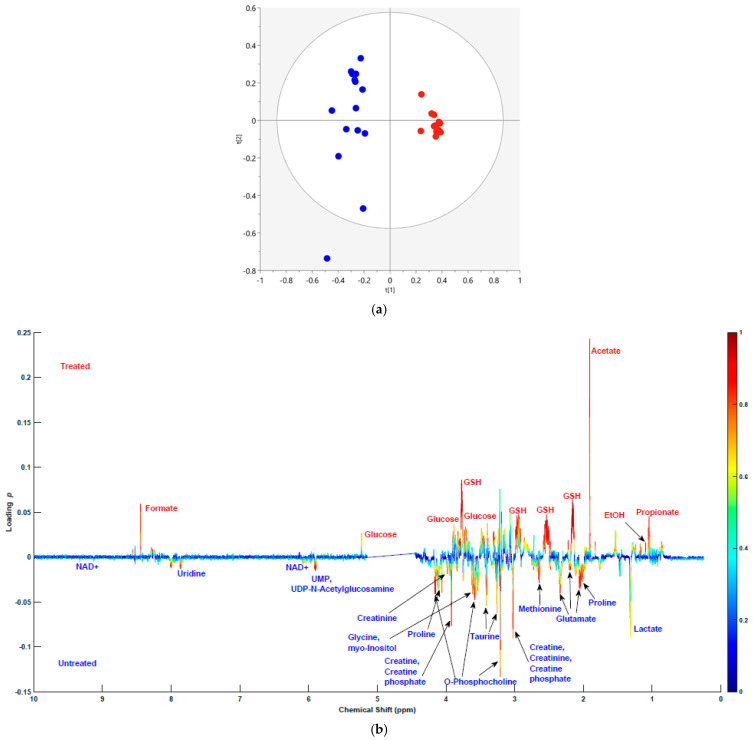
(**a**) PC1 and PC2 scores plot of the PCA analysis for the endo metabolome of 6-OHDA treated (red) and untreated (blue) cells. (**b**) Bivariate loadings plot of PC1 for the endo metabolome. The x-axis represents chemical shifts of metabolites on a parts per million (ppm) scale. The y-axis represents loading *p*-values. Overlayed are the absolute values of *p*(*corr*) as a heatmap ranging from 0 (blue) being not correlated to 1 (red) being highly correlated.

**Figure 2 molecules-30-03352-f002:**
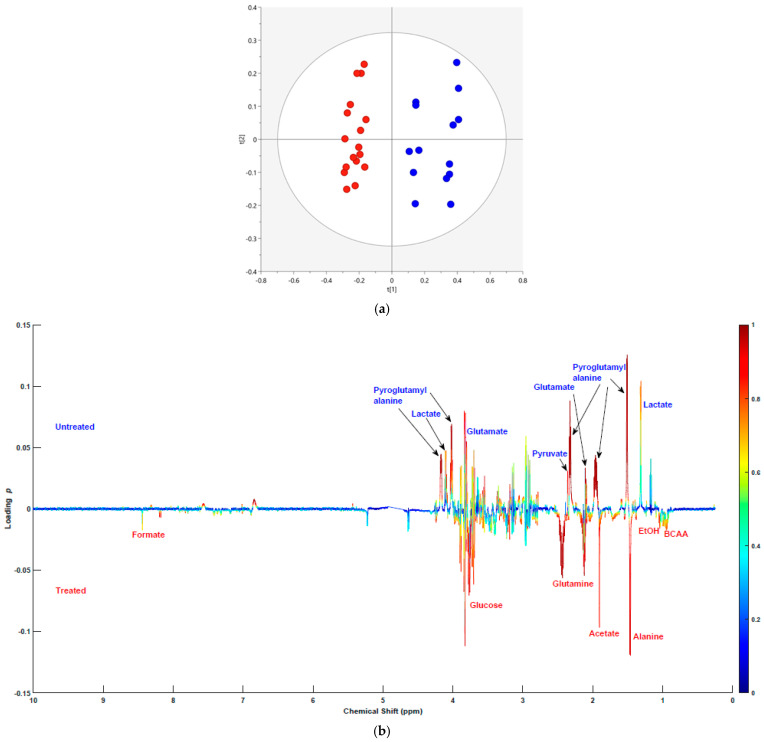
(**a**) PC1 and PC2 scores plot of the PCA analysis for the exo metabolome of 6-OHDA-treated (red) and untreated (blue) groups. (**b**) PCA bivariate loadings plot of PC1. The x-axis represents chemical shifts of metabolites on a parts per million (ppm) scale. The y-axis represents loading *p*-values. Overlayed are the absolute values of *p*(*corr*) as a heatmap ranging from 0 (blue) being not correlated to 1 (red) being highly correlated.

**Figure 3 molecules-30-03352-f003:**
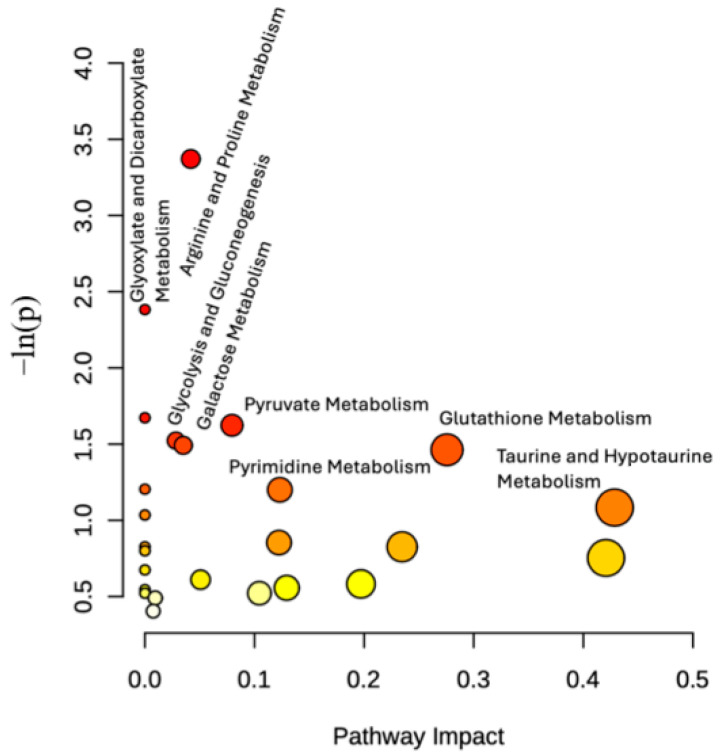
Impacted pathways in the endo metabolome. The x-axis represents pathway impact, and the y-axis pathway enrichment. Darker circle colors indicate more significant changes of metabolites in the corresponding pathway. The size of the circle corresponds to the pathway impact score and is correlated with the centrality of the involved metabolites.

**Figure 4 molecules-30-03352-f004:**
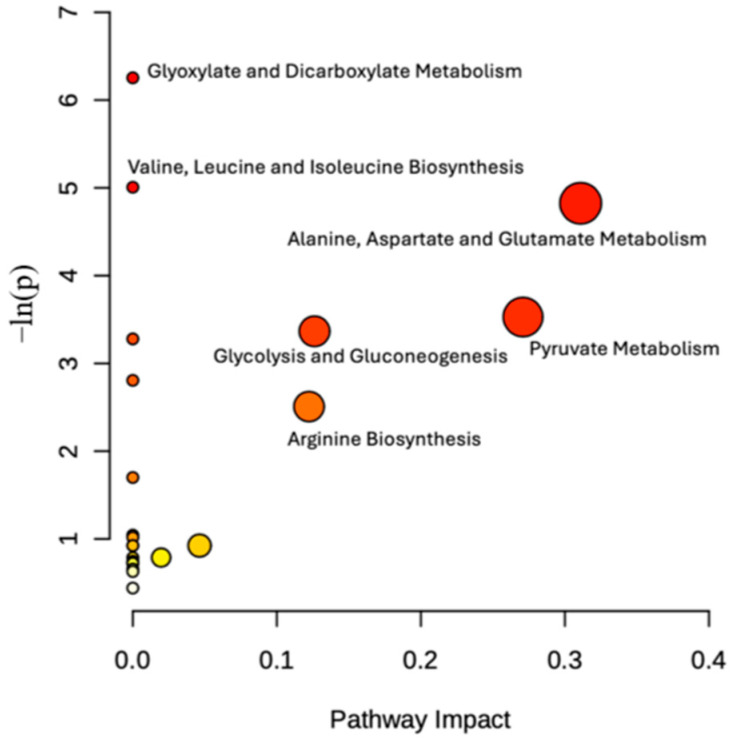
Impacted pathway in the exo metabolome. The x-axis represents pathway impact, and the y-axis pathway enrichment. Darker circle colors indicate more significant changes of metabolites in the corresponding pathway. The size of the circle corresponds to the pathway impact score and is correlated with the centrality of the involved metabolites.

**Figure 5 molecules-30-03352-f005:**
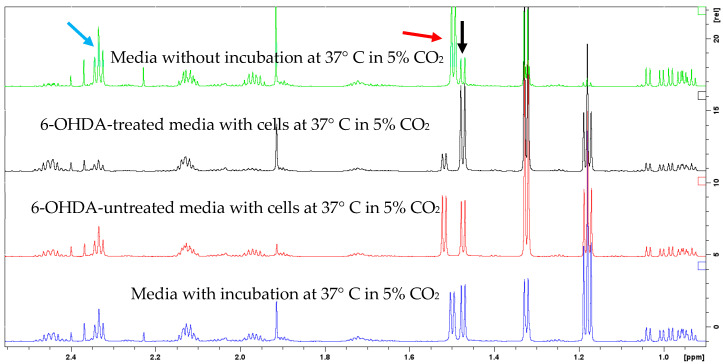
Control experiment to understand the presence of pyroglutamyl alanine. Black arrow alanine (1.49 ppm), Red arrow pyroglutamyl alanine (1.52 ppm), Blue arrow glutamate (2.33 ppm).

**Figure 6 molecules-30-03352-f006:**
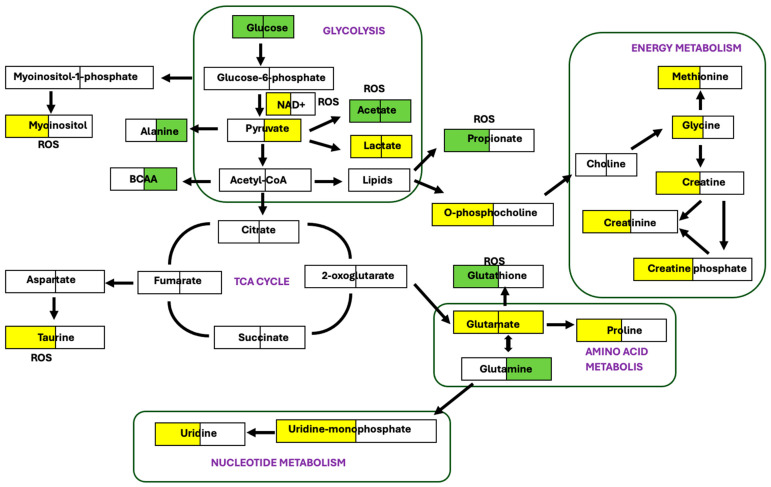
Metabolic pathways of significantly altered metabolites in cells treated with 6-OHDA compared to controls. The left side of each cell refers to the endo metabolome and the right side of the cell refers to the exo metabolome. Green colour denotes higher levels; yellow denotes lower levels. ROS: Reactive Oxygen Species.

**Figure 7 molecules-30-03352-f007:**
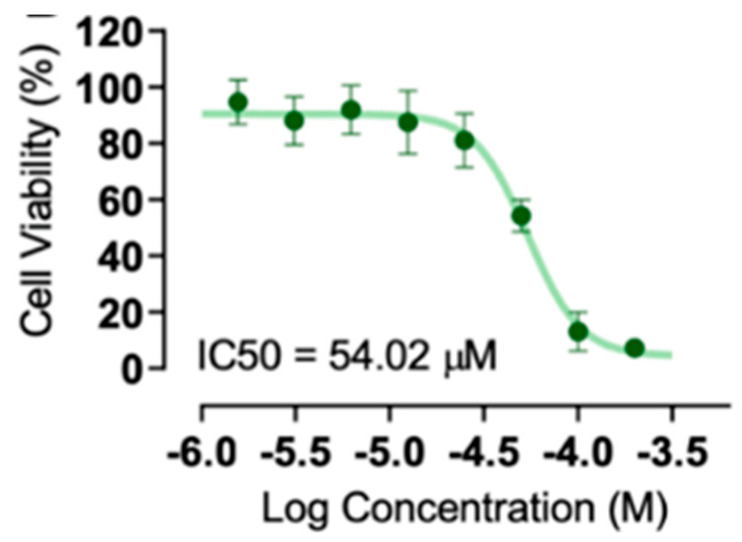
IC_50_ of 6-OHDA. The x-axis represents log concentration in molar and the y-axis represents cell viability in percent.

**Table 1 molecules-30-03352-t001:** Quantification of endo metabolites in the 6-OHDA treated group, normalised to the total metabolite content of viable cells. ** *p*-value ≤ 0.01, *** *p*-value ≤ 0.001, **** *p*-value ≤ 0.0001.

Metabolite	Unpaired *t*-Test *p*-Value (<0.05)	Mean Treated (A.U) ± SD	Mean Untreated (A.U) ± SD	Fold Change	Trend (D = Decrease) (I = Increase)
Acetate	0.0001 (***)	3.35 × 10^7^ ± 2.25 × 10^7^	1.65 × 10^7^ ± 2.42 × 10^7^	2.03	I
Creatine	<0.0001 (****)	5.16 × 10^6^ ± 4.93 × 10^6^	2.14 × 10^7^ ± 5.30 × 10^6^	0.24	D
Creatine phosphate	<0.0001 (****)	4.43 × 10^6^ ± 5.60 × 10^6^	2.40 × 10^7^ ± 5.93 × 10^6^	0.18	D
Creatinine	<0.0001 (****)	3.69 × 10^6^ ± 4.22 × 10^6^	2.07 × 10^7^ ± 4.01 × 10^6^	0.18	D
Formate	<0.0001 (****)	2.48 × 10^6^ ± 1.28 × 10^6^	1.56 × 10^6^ ± 1.38 × 10^6^	1.58	I
Glucose	0.0089 (**)	1.93 × 10^6^ ± 2.74 × 10^6^	2.66 × 10^6^ ± 7.63 × 10^6^	0.72	I
Glutamate	<0.0001 (****)	1.58 × 10^7^ ± 8.08 × 10^6^	3.67 × 10^7^ ± 8.69 × 10^6^	0.43	D
Glutathione	0.0001 (***)	4.25 × 10^6^ ± 1.22 × 10^6^	2.65 × 10^6^ ± 1.31 × 10^6^	1.60	I
Glycine	0.0001 (***)	2.40 × 10^6^ ± 1.38 × 10^6^	7.24 × 10^6^ ± 1.48 × 10^6^	0.33	D
Lactate	0.0001 (***)	4.38 × 10^7^ ± 3.59 × 10^7^	1.06 × 10^8^ ± 3.86 × 10^7^	0.41	D
Methionine	0.0026 (**)	7.33 × 10^6^ ± 2.01 × 10^6^	1.96 × 10^7^ ± 3.21 × 10^6^	0.37	D
Myoinositol	0.0001 (***)	7.93 × 10^6^ ± 5.37 × 10^6^	1.87 × 10^7^ ± 5.77 × 10^6^	0.42	D
NAD^+^	0.0001 (***)	2.62 × 10^5^ ± 1.18 × 10^5^	5.31 × 10^5^ ± 1.26 × 10^5^	0.49	D
o-Phosphocholine	<0.0001 (****)	8.41 × 10^6^ ± 4.28 × 10^6^	2.16 × 10^7^ ± 4.61 × 10^6^	0.39	D
Proline	0.0021 (**)	1.32 × 10^6^ ± 2.55 × 10^6^	2.37 × 10^6^ ± 3.35 × 10^6^	0.55	D
Propionate	0.0008 (***)	3.25 × 10^6^ ± 1.75 × 10^6^	2.44 × 10^6^ ± 1.88 × 10^6^	1.33	I
Taurine	0.0001 (***)	1.27 × 10^7^ ± 5.85 × 10^6^	2.89 × 10^7^ ± 6.29 × 10^6^	0.44	D
Uridine	0.0001 (***)	1.07 × 10^5^ ± 2.36 × 10^5^	4.95 × 10^5^ ± 2.54 × 10^5^	0.21	D
Uridine monophosphate	0.0014 (**)	2.36 × 10^6^ ± 9.97 × 10^5^	4.16 × 10^6^ ± 1.07 × 10^6^	0.56	D

**Table 2 molecules-30-03352-t002:** Quantification of exo metabolites in the 6-OHDA-treated group. * *p*-value ≤ 0.05, ** *p*-value ≤ 0.01, *** *p*-value ≤ 0.001, **** *p*-value ≤ 0.0001.

Metabolite	Unpaired *t*-Test *p*-Value (<0.05)	Mean Treated (A.U) ± SD	Mean Untreated (A.U) ± SD	Fold Change	Trend (D = Decrease) (I = Increase)
Acetate	<0.0001 (****)	1.93 × 10^7^ ± 1.88 × 10^7^	1.54 × 10^7^ ± 1.82 × 10^7^	1.25	I
Alanine	0.0001 (***)	6.39 × 10^7^ ± 1.63 × 10^8^	6.01 × 10^7^ ± 1.57 × 10^8^	1.06	I
Formate	0.0239 (*)	1.22 × 10^6^ ± 3.60 × 10^6^	1.59 × 10^6^ ± 3.49 × 10^6^	0.76	I
Glutamate	0.0001 (***)	6.50 × 10^6^ ± 7.85 × 10^7^	2.66 × 10^6^ ± 7.60 × 10^7^	2.44	D
Glutamine	<0.0001 (****)	3.51 × 10^7^ ± 8.44 × 10^7^	3.08 × 10^7^ ± 8.17 × 10^7^	1.14	I
Isoleucine	0.0001 (***)	5.24 × 10^6^ ± 1.08 × 10^7^	5.74 × 10^6^ ± 1.05 × 10^7^	0.91	I
Leucine	0.0001 (***)	6.64 × 10^6^ ± 2.78 × 10^7^	7.85 × 10^6^ ± 2.69 × 10^7^	0.84	I
Lactate	0.0001 (***)	1.37 × 10^8^ ± 2.92 × 10^8^	1.63 × 10^8^ ± 2.82 × 10^8^	0.84	D
Pyruvate	0.0001 (***)	3.54 × 10^7^ ± 6.55 × 10^6^	3.76 × 10^7^ ± 6.33 × 10^6^	0.94	D
Pyroglutamyl alanine	<0.0001 (****)	1.17 × 10^7^ ± 5.61 × 10^7^	2.13 × 10^7^ ± 5.43 × 10^7^	0.545	D
Valine	0.0099 (**)	3.38 × 10^7^ ± 1.19 × 10^8^	8.00 × 10^7^ ± 1.15 × 10^8^	0.42	I

## Data Availability

The original contributions presented in this study are included in the article/Appendix A. Further inquiries can be directed to the corresponding author.

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
