# Peer review of "Mode of Action of Toxin 6-Hydroxydopamine in SH-SY5Y Using NMR Metabolomics"

_molecules, 2025, doi:10.3390/molecules30163352_

Round 1

Reviewer 1 Report

Comments and Suggestions for Authors

The work “mode of action of toxin 6-hydroxydopamine using NMR metabolomics” by Roktima Tamuli and colleagues describes the metabolic changes observed in the SH-SY5Y cell line upon treatment with 6-OHDA. It is a detailed and well described study; however, there are a few aspects that should be improved before it can be acceptable for publication.

Major

  • Authors state that samples (n=18) originate from 3 different passages and each passage is repeated 6 times (lines 235-238). However, they are treated as if they were all independent and not biological replicates.
  • Section 2.1, table 1. Also, section 4.9 (line 326). The measures were done using raw data? Normalized? Why not use Chenomx (line 332) to provide absolute metabolite concentrations? Maybe normalized to number of living cells? Total protein content? Or even to total creatine?
  • Treated samples were treated with 6-OHDA at a concentration of 60 µM, close to IC50 (section 4.2). How did authors correct for the effect of cell death in the experiments?
  • Conclusion section, line 342 states “Depletion of ATP leads to …”. ATP was not measured (or no differences were fund in ATP levels), see table 1. This conclusion is untenable.

Minor

  • Figure 1, letters a) and b) do not appear in the figure
  • Please check the number of tabs at the beginning of each paragraph
  • Figure 3, bottom spectra, please note that the phase in the spectra is not correct, and it may affect downstream processing.
  • Line 140, please note that it is NADPH not NADH that reduces glutathione.
  • Line 295, please change “frequency” for “equivalent to”
  • Lines 290, 296 please check for extra spaces.
  • Line 299, please write in past tense
  • References section, please remove “From NLM Medline”

Reviewer 2 Report

Comments and Suggestions for Authors

This study employed NMR-based metabolomics to investigate the mode of action of 6-hydroxydopamine (6-OHDA) toxicity in the SH-SY5Y neuroblastoma cell model. The authors apply a NMR based approach on both the cell pellet (endo-metabolome) and cell medium (exo-metabolome).

The English in the manuscript needs to be revised by a native speaker. There are numerous grammatical and typographical errors, some of which are highlighted in the "Specific Comments" section below.

Due to all the concerns outlined below, I must recommend rejection of the manuscript in its current form.

Major Comments:

- Although an MTT assay was performed to determine the appropriate concentration of 6-OHDA for treatment, in my opinion, the study would have been more comprehensive if multiple concentrations of 6-OHDA had been tested and analyzed using the metabolomic approach. Therefore, my first suggestion to the authors is to include a section discussing the limitations of this type of study, along with future perspectives and potential implications, in the conclusion of the manuscript.

- The title should be more specific. I recommend including the name of the cell line, as the study does not provide a universal analysis of the mechanism of action (MoA) of this toxin.

- The abstract currently reports only the results of the study. Please revise the abstract accordingly. I recommend following the Molecules Guide for Authors, which provides clear guidelines on how to structure the abstract.

Abstract: The abstract should be a total of about 200 words maximum. The abstract should be a single paragraph and should follow the style of structured abstracts, but without headings: 1) Background: Place the question addressed in a broad context and highlight the purpose of the study; 2) Methods: Describe briefly the main methods or treatments applied. Include any relevant preregistration numbers, and species and strains of any animals used; 3) Results: Summarize the article's main findings; and 4) Conclusion: Indicate the main conclusions or interpretations. The abstract should be an objective representation of the article: it must not contain results which are not presented and substantiated in the main text and should not exaggerate the main conclusions.

In particular, it is important to include background information (i.e., what 6-OHDA is and its context of use) as well as the methodology (specifically, the exo- and endo-metabolomic approach).

- The introduction section needs to be substantially expanded. Here are some specific suggestions:

- Expand the context: How is 6-OHDA used, in what types of studies, and for what purposes?

- Why is it important to study the mechanism of action (MoA) of 6-OHDA toxicity? Please provide a broader context supported by relevant literature. Is its mechanism not fully understood? Why is there a need for this study in the current literature?

- Lines 31–32: Why was NMR metabolomics chosen? Please cite relevant studies that have used various approaches—both chemical and biological—for investigating the MoA of 6-OHDA.

- Lines 37–40: While the importance of NMR metabolomics is introduced, please add a paragraph specifically describing the two approaches used in this study: endo- and exo-metabolomics.

- Lines 42–43: These lines should be removed, as this is not the appropriate place for a conclusion.

- The identification of pyroglutamyl alanine is not clearly explained. Please include the 2D NMR experiments that support the reported correlations.

- Figure 3a is not only a structure of the molecule. Please revise the caption to reflect its full content.

- Figure 3b is not of significant interest in its current form. The phrase “blue bold” is unclear—did you mean the arrows? The type of correlation shown in the figure (e.g., HMBC or TOCSY) is standard. You must provide the actual 2D NMR spectra that demonstrate this correlation. Additionally, please cite literature that confirms the presence of this metabolite in the medium.

- The statement in lines 205–206 does not describe a real limitation—it is a result. Similarly, the other points mentioned are not true limitations. A real limitation is discussed in the following comments.

- To identify altered metabolic pathways based on the identified metabolites, you should use an appropriate tool such as MetaboAnalyst.

Specific comments:

- Please adjust Figure 1A by grouping the two panels and adding the letters “a” and “b.”

- Line 66: Please correct the spelling of “heatmap.”

- Figures 1 and 2: Please add the phrase “of explained total variance by PC1 and PC2 of the PCA analysis.” Also, include this information in the text.

- Adjust Figure 3: The text on the right side of the figure is difficult to read and hard to correlate with the different spectra. Please show only an expanded view of the region of interest. It is not necessary to include the 2.5–4.5 ppm region.

- Line 265: Add information about the instrument used for lyophilization.

- Line 290: Clarify how many samples were used for the J-resolved experiment. Also, correct the grammatical error in the sentence.

- Lines 291–292: For the HSQC-TOCSY experiment, please correct the error (also present at line 335). Clarify how many samples were used for this 2D experiment (one PQC sample or more?).

- Line 295: Please correct the term “Fourier Transform.”

- Line 298: Correct the word “spectr.”

- I suggest exchanging the order of paragraphs 4.9 and 4.10, since metabolite identification was likely done first, followed by integration and quantification.

- Change the title of the paragraph “Statistical analysis” to “Metabolite integration and statistical analysis.”

- Lines 344–345: These two lines may be misleading. It is clear from the introduction that NMR metabolomics is useful for studying this type of process. Instead of these lines, add some future perspectives related to this study and its implications connected to 6-OHDA.

- Figures S1 and S2 have the same caption. Please correct this.

Comments on the Quality of English Language

The English in the manuscript needs to be revised by a native speaker. There are numerous grammatical and typographical errors, some of which are highlighted in the "Specific Comments" section below.

Round 2

Reviewer 1 Report

Comments and Suggestions for Authors

The revised version of the work “mode of action of toxin 6-hydroxydopamine using NMR metabolomics” by Roktima Tamuli and colleagues has addressed most of the concerns by this reviewer, however, there is one major concern remaining.

Major

As the authors acknowledge in their response, they consider technical replicates as biological in order to increase the sample size. This may lead to overestimation of differences and wrong conclusions.

Reviewers proposal: Repeat one biological experiment with 3 replicates, as done before. If no differences between the new and old data are detected (the 6 new samples, 3 case, 3 controls, cluster with the ones in the manuscript) this will validate the entire dataset.

Other

In table 1 please add statistics corrected for multiple comparisons, t-test alone is not sufficient.

Author Response

Please see attached response.

Reviewer 2 Report

Comments and Suggestions for Authors

The authors have addressed all the comments, and the article can be accepted in its current form.

Author Response

No revision was requested by the reviewer.